# The Use of Antibiotics and Antimicrobial Resistance in Veterinary Medicine, a Complex Phenomenon: A Narrative Review

**DOI:** 10.3390/antibiotics12030487

**Published:** 2023-03-01

**Authors:** Alice Caneschi, Anisa Bardhi, Andrea Barbarossa, Anna Zaghini

**Affiliations:** Department of Veterinary Medical Sciences, Alma Mater Studiorum—University of Bologna, Via Tolara di Sopra 50, Ozzano dell’Emilia, 40064 Bologna, Italy

**Keywords:** veterinary medicine, responsible and prudent use of antibiotics, PK/PD approach, antibiotic-resistance

## Abstract

As warned by Sir Alexander Fleming in his Nobel Prize address: *“the use of antimicrobials can, and will, lead to resistance”*. Antimicrobial resistance (AMR) has recently increased due to the overuse and misuse of antibiotics, and their use in animals (food-producing and companion) has also resulted in the selection and transmission of resistant bacteria. The epidemiology of resistance is complex, and factors other than the overall quantity of antibiotics consumed may influence it. Nowadays, AMR has a serious impact on society, both economically and in terms of healthcare. This narrative review aimed to provide a scenario of the state of the AMR phenomenon in veterinary medicine related to the use of antibiotics in different animal species; the impact that it can have on animals, as well as humans and the environment, was considered. Providing some particular instances, the authors tried to explain the vastness of the phenomenon of AMR in veterinary medicine due to many and diverse aspects that cannot always be controlled. The veterinarian is the main reference point here and has a high responsibility towards the human–animal–environment triad. Sharing such a burden with human medicine and cooperating together for the same purpose (fighting and containing AMR) represents an effective example of the application of the One Health approach.

## 1. Introduction

In the present narrative review, the terms “antibiotic” and “antibacterial” are used instead of “antimicrobial” because they are specifically related to bacteria, which are the microorganisms considered in the review. Moreover, the term “antibiotic” and its derivations (e.g., antibiotic resistance indicative of antimicrobial resistance) are generally accepted.

The discovery of antibiotics in 1928 revolutionized modern medicine, as they were deemed “miracle drugs”. In addition to their use to fight infectious (bacterial) diseases, antibiotics have enabled remarkable medical improvements. Without them, major surgeries, organ transplantation, or cancer chemotherapy, would be impossible [1]. The availability of antibiotics resulted in a substantial improvement of the health and well-being of animals as well [2,3]. Furthermore, the change in livestock production systems (from extensive livestock production to intensive industrialized production) coincided with the early use of antibiotics in food-producing animals [4,5].

However, as warned by Sir Alexander Fleming in his Nobel Prize address, *“the use of antimicrobials can, and will, lead to resistance”* [6]. Bacteria are simple and resilient microorganisms capable of changing themselves when the conditions of their environment are no longer advantageous, for example, in the presence of antibiotics. Therefore, antimicrobial resistance (AMR) has to be considered a natural phenomenon to which any antibiotic contributes. The AMR threat is mainly due to the overuse and misuse of antibiotics, often unnecessarily or without a prescription. Additionally, the use of antibiotics in animals (food-producing, companion, or exotic) has significantly contributed to the selection and transmission of resistant bacteria [7,8].

As this review demonstrates, the epidemiology of resistance is complex, and factors other than the overall quantity of antibiotics consumed may influence the level of AMR. The horizontal spread of antibiotic resistance genes (ARGs), inadequate sanitation, movements of humans, and the trade of live animals and meat are just some examples [9].

Globally, the threat and costs of AMR are widely recognized. For example, antibiotic-resistant pathogens cause approximately 33,000 deaths per year in Europe and 4.95 million total deaths worldwide, resulting in EUR 1.5 billion in healthcare expenses and productivity losses annually [10]. The World Health Organization (WHO) considers AMR *“one of the ten global public health threats facing humanity”* based on the predictions that, by 2050, it will be responsible for more than 10 million deaths per year, with a loss of more than 100 trillion dollars worldwide [11,12].

In this context, to raise awareness about the AMR phenomenon and the appropriate use of antibiotics in animals, the European Union, as well as worldwide public or private organizations, highlighted in Regulation (EU) 2019/6 of the European Parliament and of the Council that its cross-border dimension, economic impact, and animal and human health consequences requires global intersectoral action to combat AMR [13].

Veterinary medicine is concerned with the care and well-being of animals and, indirectly, that of humans. It should be noted that the term “animals” is very generic, encompassing terrestrial, aquatic, and avian species, which can be further divided into production, companion, and exotic/wild animals, all sharing different ecosystems with humans, directly or indirectly. These animals have many diseases in common with humans, even infectious ones (e.g., bacterial, viral, parasitic), although they are usually caused by different etiological agents. Therefore, it is also common to share medications, including antibiotics. As underlined by Regulation (EU) 2019/6 [13], the AMR phenomenon in animals is much more complex than it is in humans and requires an even more attentive and conscious use of antibiotics. In recent years, the application of the pharmacokinetics/pharmacodynamics (PK/PD) approach to define the correct dosage regimen of antibiotics has represented an important tool in veterinary medicine, improving the patient outcome and limiting the selection of resistant mutants. As is described in more detail later (see Section 4), the PK/PD model is viable “to translate” the antibacterial activity into clinical situations, ensuring a successful antibiotic outcome. The PK/PD indices (i.e., AUC/MIC, Cmax/MIC, or %T>MIC) combine pharmacokinetics (describing the time course of the concentration of a drug in the blood) and pharmacodynamics (correlating the intensity of the effect of a drug to its concentration), allowing for the prediction of the variation of the effect over time.

However, despite the acknowledged importance of the PK/PD approach and the responsible and prudent use of antibiotics, there are still many diverse aspects that remain uncontrollable (e.g., interspecies differences in kinetics and dynamics are just one of many possible examples).

Focusing on the use of antibiotics in different animal species and its impact on animals, but also on humans and the environment, the purpose of this narrative review is to provide an overview of the AMR phenomenon in veterinary medicine, trying to explain its complexity.

The literature is filled with valuable contributions to these topics (e.g., systematic or narrative reviews, articles, research), and given the nature of review (narrative review), only some particular examples are provided here to better understand the vastness of the issue. Finally, the authors’ intent was to address not only researchers and scientists, but also everyone involved (e.g., human and veterinary practitioners, veterinary associations, pharmaceutical industry) in containing this *“global public health threat facing humanity”*, as defined by the WHO.

## 2. Antimicrobial Resistance (AMR) in Humans and in Animals

Antimicrobial resistance (AMR) is a natural phenomenon that appears when a bacterium (pathogen or commensal) susceptible to an antibiotic, following its exposure to that antibiotic, becomes unsusceptible (acquired resistance).

The consistent usage of antibiotics (e.g., prescribed for therapy, misused, or overused) creates a selective pressure on pathogenic bacteria but also on commensal bacteria (e.g., gut or skin microbiota) that can determine the prevalence of antibiotic-resistant bacteria (ARBs) and the spread of resistant bacteria or genes. Plasmids, bacteriophages, or integrative mobile genetic elements facilitate horizontal gene transfer; therefore, it is essential to reduce and to limit the use of antibiotics by following the “principles of responsible and prudent use” in order to preserve their efficacy [14,15,16].

It is important to keep in mind that AMR is a dynamic process: once bacteria become resistant, redeveloping susceptibility to antibiotics is not guaranteed; it takes time and only occurs when selection pressure decreases [17]. In a bacterial population, some bacteria may carry ARGs. These genes are often unable to confer a clinically relevant resistance to an antibiotic, but they contribute to reduce the susceptibility of bacteria (low-level resistance), thereby promoting the selection of mutants with a higher level of resistance, especially in the presence of low concentrations of the drug. Low-level resistance is often the first step in the development of high-level resistance [18]. Finally, when a resistant population replaces a population of susceptible wild bacteria in the environment “at low cost to adaptation”, it becomes very stable in its ecosystem [19]. However, evidence has shown that the prevalence of ARBs in food-producing animals and humans has decreased as a result of restrictions on the use of antimicrobials [20].

Vertical (to bacterial offspring) and horizontal (to other bacterial strains or species) ARG transfers are well known and demonstrated [21]. Due to the high density and biodiversity of microorganisms, the gut microbiota of humans, food-producing animals, and companion animals provide ideal conditions for the spread of ARGs and may represent an important reservoir for ARGs, regardless of prior antibiotic exposure [22,23]. For example, high numbers of ARGs, including those to critically important antimicrobial (CIAs) classes (e.g., quinolones and carbapenems), have been detected in manure from animals that have not received antibiotics [24]. Frequently, at the end of antibiotic treatments, ARGs can be found in the farm environment [25]. Aquatic environments such as wastewaters, hospital effluents, and animal or agricultural drainage may also represent threatening reservoirs for ARGs [18].

The transmission of resistant bacteria typically occurs from animals to humans through various routes, but it can also be of human origin, such as in the case of methicillin-resistant *Staphylococcus aureus* (MRSA) [26,27]. Three types of AMR can be observed in animals: (i) AMR in specific animal pathogens; (ii) AMR in zoonotic pathogens; and (iii) AMR in commensal bacteria. Type 3 is of particular concern from an ecological point of view due to the larger biomasses of these bacteria compared to the other groups [28].

### 2.1. Food-Producing Animals

Mostly through manure, antibiotic residues, ARBs, and ARGs are spread from food-producing animals, being of concern for the environment, wild fauna, and humans.

The most commonly used antibiotic classes in food-producing animals are tetracyclines, β-lactams, quinolones, and sulfonamides, with oral administration being the most convenient route [29,30]. Tetracyclines have low oral bioavailability in pigs (5–15%) [29,30] and poultry (≤5%) [31], as does ampicillin (10%) in pigs [32]. The unabsorbed antibiotics alter the intestinal microbiota rapidly [33] and remain microbiologically active in the feces excreted, affecting environmental microorganisms as well. In the case of AMR to tetracyclines, for example, it is generally linked with multidrug-resistant bacteria co-selecting resistant genes to CIAs for humans [34]. This may be the case for carbapenems, which are not used in food-producing animals, but resistance genes have been found in livestock, specifically in pigs and chickens [35,36].

As previously mentioned, the land application of food-producing animal manure containing resistant bacteria and antibiotic residues may create an environmental reservoir of ARBs [24,37,38], potentially leading to further dissemination, for example, into wild animal populations [39].

Furthermore, the abundance of ARGs in the feces of people working in pig/broiler farms has often been linked to livestock AMR, potentially due to exposure to animal feces and dust in farms. This suggests that antibiotic residues in farm dust may also impact the gut microbiota by selecting for resistance [40]. The food chain (milk, eggs, and especially meat) represents one of the most important sources of ARB and ARG transmission between animals and humans [41].

As is discussed in more detail later, aquatic environments may also become reservoirs of ARBs and ARGs from various sources, such as wastewaters, hospital effluents, agricultural contamination from animals, and aquaculture. Antibiotics in aquaculture are mainly administered through feed, immersion, or direct application into water [42,43], with 70–80% being dispersed into water systems [44].

### 2.2. Companion Animals

Antibiotic-resistant bacteria and antibiotic resistance genes, as well as multidrug-resistant bacteria (MDR), have emerged and spread among dogs and cats (Table 1).

Bacterial resistance to the most conventional antibiotics licensed for veterinary use poses a serious risk for animals and for humans. For example, Methicillin-resistant *S. pseudintermedius*, (MRSP)/methicillin-resistant *S. aureus* (MRSA), and MDR Gram-negatives are hospital-associated pathogens that are currently spreading outside of healthcare facilities with high transmission ability, representing a serious risk to human health, especially for frail subjects (e.g., children, elderly, subjects with compromised immune systems) [54].

Moreover, the tight contact between companion animals and humans (e.g., petting, licking, or physical injuries) as well as the home environment facilitate the transmission of antibiotic-resistant bacteria and genes [55].

Zhou et al. [56] conducted a study investigating the prevalence of drug resistance phenotypes and ARGs of ESBL *E. coli* in dogs and cats in China. *E. coli* is one of the most common microorganisms in the intestinal flora and is continuously exposed to antibiotic selection pressure. The authors found a prevalence of antibiotic-resistant *E. coli* exhibiting a variety of antibiotic resistance patterns equivalent to 86.64% (higher than results from other studies) [57,58,59,60,61,62,63]. Moreover, the study found that ESBL *E. coli* had a higher prevalence in cats than in dogs, which differs from the findings of other research [64]. Zhou et al. [56] hypothesized that this may be due to the higher use of third-generation cephalosporins in cats compared to dogs. Cats are less likely to take oral medication or medication in food, leading to a preference for injectable preparations [56,65]. *E. coli* isolated from dogs and cats is highly similar to human isolates, increasing the potential for transferring resistant strains between species [66].

As previously mentioned, Tóth et al. [16] underlined the closeness between humans and companion animals. The study conducted by the authors aimed to (i) detect the ARGs’ content of canine saliva, (ii) attach the ARGs with the bacterial species of origin, and (iii) define the ARGs’ spreading capacity. The species *Bacteroides* spp., *Capnocytophaga* spp., *Corynebacterium* spp., *Fusobacterium* spp., *Pasteurella* spp., *Porphyromonas* spp., *Staphylococcus* spp., and *Streptococcus* spp. were considered the most common pathogens in dog bites. By mapping the genome of the dog saliva bacteria, the authors detected the genomes of the most relevant bacteria in dog bite infections, as well as the genes conferring resistance to aminoglycosides, carbapenems, cephalosporins, glycylcyclines, lincosamides, macrolides, oxazolidinones, penams, phenicols, pleuromutilins, streptogramins, sulfonamides, and tetracyclines. The identified antibiotic resistant genes have the capacity to establish themselves in the human body and bacteriota.

## 3. Antibiotics in Veterinary Medicine: Responsible and Prudent Use

The most frequently used classes of antibiotics in animals (food-producing, companion, and exotic animals) are quinolones (especially fluoroquinolones), aminopenicillins alone or in combination with potentiators, first- and second-generation cephalosporins, tetracyclines, sulfonamides alone or in combination with potentiators, and also third- and fourth-generation cephalosporins, macrolides, and glycopeptides, categorized as highest priority critically important antibiotics (HPCIAs) [14,67].

The “Terrestrial Animal Code”, issued by the Office International des Epizooties (OIE) in 2022 [68], represents recent guidance for the responsible and prudent use of antimicrobials in veterinary medicine. It settles the respective responsibilities of the competent authority and stakeholders in all processes concerning veterinary drugs (i.e., antibiotics) ranging from authorization to the administration to the animals.

In particular, the authorization process for new veterinary medicinal products containing antibiotics focuses on three key points for the responsible and prudent use of such drugs: (A) the *“Assessment of the potential of antimicrobial agents to select for resistance”*, consisting of *(i) the concentration of either active antimicrobial agents or metabolites in the gut of the animal (where the majority of potential foodborne pathogenic agents reside) at the defined dosage level, (ii) the pathway for human exposure to antimicrobial-resistant microorganisms, (iii) the degree of cross-resistance, and (iv) the intrinsic and pre-existing baseline level of resistance in the pathogenic agents of human health concern;* (B) the *“Establishment of acceptable daily intake (ADI), maximum residue limit (MRL) and withdrawal periods in food-producing animals”*, concerning the safety evaluation, which should include the potential biological effects of the antibiotic residue on the intestinal biota of humans, referring to the values of minimum selective concentrations (MSCs; MSC*_denovo_* and MSC*_select_*); (C) the *“Post-marketing antimicrobial surveillance”*, since, among the information underpinning the pharmacovigilance programs, the lack of efficacy is of great concern to minimize AMR [68].

Access to antibiotics, especially in the livestock sector, varies significantly around the world and sometimes within a single country, particularly in low-income countries. For instance, in many parts of Africa, there is evidence of the use of substandard and non-registered drugs. These antibiotics may be falsely branded or expired, therefore containing a lower (or no) amount of the active ingredient than indicated or even containing harmful ingredients [15]. Geographical differences concerning drug distribution channels are an additional factor; in many low- and middle-income countries, it is possible to buy antibiotics over the counter (OTC) without a prescription. Furthermore, the trade of antibiotics over the internet represents a new, unregulated source of these drugs [15].

In all situations where the administration of an antibiotic in animals is really necessary, the principles for its responsible and prudent use must be followed, including (i) using quality-assured products; (ii) avoiding the regular preventive use of antibiotics and, in livestock, their use as growth promoters; (iii) avoiding using HPCIAs for human medicine; (iv) only using antibiotics based on the diagnosis of disease (to know the pathogen causing the infection) by a veterinarian and only for the authorized indications; (v) in livestock, promoting the treatment of individual animals with the correct dose and duration, avoiding group treatments, especially through feed; (vi) whenever possible, selecting antibiotics for therapeutic use based on antibiotic susceptibility testing [15]. It is important to note that, particularly in food-producing animals, these principles can only be applied if proper species-specific welfare conditions are also provided to animals, including the implementation of appropriate external and internal biosecurity measures and vaccination schemes [15,69].

Other important factors to consider when setting up a correct antibiotic treatment in animals include (i) antibiotic use only when a primary bacterial infection has been proven; (ii) the severity of the infection and its likelihood to resolve without the use of antibiotics; (iii) especially in case of empirical prescription, knowing the spectrum of antibiotics (narrow spectrum antibiotics are to be preferred), their pharmacokinetics and pharmacodynamics, and potential toxicity or adverse effects; (iv) defining the most suitable route of administration (local treatment is to be preferred over a systemic one) [14].

The need to reduce the use of antibiotics in animals puts a great emphasis on the prescriptions of antibiotics by veterinarians to ensure their responsible and prudent use. Veterinarians can prescribe or dispense antibiotics based on defined criteria (Table 2).

Several studies have analyzed prescriptions of antibiotics for companion animals in veterinary university hospitals and private veterinary practices in different countries [70,71,72,73,74,75]. Beta-lactams (penicillins and cephalosporins) resulted in the most commonly prescribed family of antibiotics for companion animals, along with cotrimoxazole, tetracyclines, fluoroquinolones, and macrolides. According to responses to a questionnaire on the rationale for the use of CIAs, the lack of conventional antibiotics (non-CIAs) for pets or authorized suitable pharmaceutical dosage forms were often cited as the reason for the use of CIAs (70% in 2019) [17].

Antimicrobial stewardship focuses on the approaches necessary to reduce the resistance selection and to preserve the clinical effectiveness of antibiotics, implying a wider concept than “prudent use” or “responsible use”. Good stewardship practices (GSP) is a guiding principle based on the active and dynamic process of the continuous improvement of antibiotic use that should be followed by everyone involved. The main actors are veterinary practitioners, but laboratory diagnosticians, owners, drug regulators, and pharmaceutical companies are relevant stakeholders [76].

### 3.1. Food-Producing Animals

Antibiotics are administered to food-producing animals for several purposes, including therapeutic, metaphylactic, and prophylactic use, but they are also used as growth promoters. The latter involves administering low doses (sub-therapeutic) of antibiotics in feed or water over an extended period of time to improve growth and production efficiency [77]. Despite being prohibited as growth promoters in most countries, antibiotics are still used extensively for this purpose in certain regions/countries, including India and China [78,79]. In 2016, Vishnuraj et al. [80] highlighted a global average annual consumption of antibiotics >100 mg/kg of animal produced, which is expected to grow by 2030. Similarly, Van Boeckel et al. [81] evaluated that the global average annual consumption of antibiotics was approximately 45 mg/kg of animal produced in cattle, 148 mg/kg in chickens, and 172 mg/kg in pigs, with a significantly growing prediction by 2030, particularly in some countries (Brazil, Russia, India, China, and South Africa) [81].

The above previsions were confirmed by Tiseo et al. [82]. The authors estimated that antimicrobial use in chicken, cattle, and pigs was 93,309 tonnes of active ingredient in 2017, projecting an increase of 11.5% by 2030 to 104,079, with different contributions related to the considered animal species. In both 2017 and 2030, Asia consumed the most significant amounts of antibiotics, with an expected increase of 10.3% (the foreseen Asian antibiotic use in 2030 amounts to 68% of the antibiotics used worldwide in 2017); Africa has the highest expected increase by 2030 (37%), while Oceania, North America, and Europe are expected to have the smallest percentage increase in antibiotic sales (3.1%, 4.3%, and 6.7%, respectively).

Concerning Europe, the trend highlighted by Tiseo et al. [82] was confirmed by the efforts made to decrease the use of antibiotics through the European Union’s “Farm to Fork Strategy,” which aims for a 50% reduction in antimicrobial sales for farmed animals and aquaculture by 2030 [77,83].

In most cases, the antibiotics available for food-producing animals are the same used in human medicine and can contribute to AMR [84,85], as well as to the spread of ARBs detected in livestock farms (e.g., poultry and pigs) among workers, animals, and the environment [86,87].

### 3.2. Companion Animals

In the therapy of pets, the extra-label use of drugs, and specifically of antibiotics, is frequent and more common than in food-producing animals [88]. The reasons behind this practice can be very different and are detailed in Section 5.3.

The regulations concerning the extra-label use of drugs show significant differences among countries; for example, while in the United States, the extra-label administration of drugs, and specifically of antibiotics, in companion species is legal, and in Europe, the extra-label use of drugs is strictly regulated [88].

In the United States (but the situation is quite similar worldwide), more than 60% of households were associated with the ownership of various pet species [89]. During the coronavirus disease (COVID-19) pandemic outbreak, a further increase was observed in the number of companion animal acquisitions [90,91]. At the same time, the awareness of owners towards their pets has changed, with most of them now viewing their pets as family members [92], often sleeping together with their owners and licking their faces. Concerning dogs, the occurrence of dog bites is currently frequent, although often under-reported [93]; however, dog attacks have been steadily rising in the United States, Europe, Canada, and Australia [94].

Nowadays, humans and animals share most antibiotic classes, as well as ARBs and ARGs [16,95]; thus, in addition to food-producing animals and their derived food, companion animals also play a crucial role in the circulation of AMR [96,97,98,99,100]. Guardabassi et al. and Pomba et al. [96,99] emphasized that it is possible for AMR fecal bacteria to be transferred from animals to humans in the same household, as well as from humans to pets. This is the case of methicillin-resistant *S. aureus* (MRSA), which is of human origin. Companion animals may be “transient carriers” of MRSA, but the source can only be from an infected human or a human carrier living in the same household. In conclusion, the authors agreed that the occurrence of MRSA in pets is most likely of human origin, with pets serving as transient carriers, but potentially affecting humans.

Despite the antibiotic use in companion animals being considerably lower than that in food-producing animals, the use of CIAs belonging to the AVOID USE and/or RESTRICT USE categories of the European Medicines Agency (EMA) classification is common in companion animals. For example, in Denmark, the majority of veterinary fluoroquinolone global consumption is intended for companion animals, despite that this represents only 1% of total veterinary antibiotics used [54,101].

## 4. The PK/PD Approach for Antibiotics in Veterinary Medicine

In both human and veterinary medicine, defining the correct dosage regimen (i.e., dose, intervals between doses, and duration of treatment) is essential to ensure the health of patients with minimal adverse effects. This is especially true for antibiotics, which also have the additional purpose of countering or suppressing antimicrobial resistance. Therefore, the “correct” dose and dosing regimen for antibiotics is of the utmost importance. Antimicrobials must reach active (i.e., therapeutic) concentrations at the site of infection, the so-called “biophase”, and remain there for a sufficient amount of time to ensure healing.

The knowledge of both pharmacokinetics and pharmacodynamics is essential, as is the definition of PK and PD parameters. The PK/PD approach, together with PK/PD indices, helps to integrate the two processes (PK and PD) and optimize the dosage regimens of drugs when systemic action is required.

Concerning antibiotics, PK/PD indices are useful for determining the dose to be administered, but they are also necessary for establishing clinical breakpoints (CBSs) that should be considered as MIC values (mg/L) or zone inhibition diameters used to categorize antibiotics tested for their antimicrobial susceptibility as susceptible (S), intermediate or susceptible-increased exposure (I), or resistant (R) [102].

PK/PD indices link antibiotic exposure to the antibiotic susceptibility of an infecting pathogen. Generally, the susceptibility is described as the minimum inhibitory concentration (MIC), which is the lowest concentration (mg/L) of an antibiotic that, under defined in vitro conditions, prevents the growth of a microorganism within a defined period (18–24 h). It is specific for a given microorganism and ultimately reflects the outcome of the antibacterial action (growth and kill/death of bacteria) [102,103].

MIC is widely accepted as a PD parameter in all PK/PD indices of antibiotics since it reflects the susceptibility of bacteria to antibiotics. Generally, for all drugs, the three parameters that describe the PD profile are efficacy, potency, and sensitivity, all of which are encompassed by MIC. Additionally, MIC is dependent on the assay condition and variability, making it a variable reflecting the three PD parameters rather than a PD parameter itself. MIC has been related to PD parameters by equations; thus, it can be regarded as a hybrid PD parameter [104,105,106].

The time–kill curve assay (TKCA) is another index describing the bacterial susceptibility to an antibiotic. It has two main purposes: (i) to categorize the antibiotic as concentration-, time-, or co-dependent; (ii) to calculate PK/PD indices, for example AUC/MIC, when the in vivo effects of the antibiotic depend on the highest achievable antibiotic concentration above the MIC for a pathogen to ensure maximal kill (e.g., aminoglycosides or quinolones). The “matrix effect” represents the major limitation of TKCA [102,107].

Common PK/PD indices include: (i) the percentage of a 24 h period that the antibiotic concentration exceeds the MIC of the pathogen (at steady-state conditions) (%T>MIC; e.g., β-lactam antibiotics); (ii) the highest concentration of the antibiotic to MIC ratio (Cmax/MIC); and (iii) the area under the antibiotic concentration-time curve over 24 h at steady-state (sometimes referred to as “internal dose”) to MIC ratio (AUC/MIC; e.g., fluoroquinolones, aminoglycosides, macrolides, tetracyclines) [102]. Both AUC/MIC and %T>MIC are ordinarily used in veterinary medicine [102].

Monte Carlo methods are stochastic computational algorithms (based on repeated random sampling) that are routinely used to support several aspects of antimicrobial PK/PD for different purposes: (i) the establishment of CBPs for antibiotic susceptibility tests [108]; (ii) the computation of PK/PD cut-offs by both CLSI [109] and VetCAST approaches [110]; (iii) the computation of an empirical population dose for antibiotics, taking into account the available MIC distributions. Given the high variability of in vitro studies and the difficulty of in vivo studies, the Monte Carlo simulation represents a valuable aid. In all PK/PD indices, the free antibiotic (unbound fraction to plasma protein) should be considered [102,103]. Drug action is mediated by the plasma concentration of the free drug and its time course; only this fraction, crossing capillary and tissue membranes, is able to achieve the biophase and bind to its target (e.g., receptor, enzyme, microorganism). In the absence of specific barriers, the free drug concentration in plasma balances with the free drug at the biophase [102]. Nevertheless, several antibiotics (e.g., fluoroquinolones: marbofloxacin, enrofloxacin, ciprofloxacin, and pradofloxacin) are characterized by biophase concentrations higher than expected in some species (pigs, calves, and dogs) [111,112,113,114]. To amend these gaps, for drugs featured by a moderate percentage of plasma protein binding (in dogs: marbofloxacin 22%; enrofloxacin 35%; and ciprofloxacin 18%, respectively), plasma (total, meaning bound and unbound) concentrations are generally a suitable “surrogate” to assess the active concentrations and to compute PK/PD indices [113].

The percentage of protein binding of a drug is species-specific, and it is susceptible to variations related to physiological or pathophysiological conditions of animals. Mzyk et al. [115] pointed out age as an additional factor affecting the percentage of protein binding. The authors evaluated the extent of the plasma protein binding of danofloxacin, florfenicol, and tulathromycin in calves. The observed main differences concern levels of albumin (lower at one day than in two-and six-month-old calves) and alpha1-acid glycoprotein (lower in calves up to 21 days of age). Concerning the three considered antibiotics, the authors indicated a high inter-individual variability but no significant age-related effects on plasma protein binding.

### 4.1. Mutational Resistance during Antibiotic Treatment

During antibiotic treatment against a pathogen bacterium, the risk of emergence and propagation of antibiotic-resistant microorganisms should be considered, even though this emergence is limited [103,116,117].

Several studies, mostly in vitro and simulating antibiotic “therapeutic” dosing, suggest that exposure associated with an increased probability of therapeutic success may be insufficient to suppress the emergence of ARBs, particularly in Gram-negative bacteria [116,117]. Despite the importance and the utility of MIC in the PK/PD approach, when indices have to be defined to suppress AMR emergence, some studies suggest using alternative parameters to measure the susceptibility of the bacteria versus antibiotics, which are able to estimate the potential for resistance [102].

The Mutant Prevention Concentration (MPC) is defined as the MIC of at least one susceptible single-step mutant. MPC measures the MIC of the most resistant sub-population that may proliferate at concentrations higher than the MIC [102,118,119]. The antibiotic concentration ranging between the MIC and the MPC is the mutant selection window (MSW), which represents the antibiotic concentration range for which the evolution of resistance can occur by selecting for the non-susceptible portion of the population. The MSW changes and widens as resistance evolves. To prevent the selection of the first mutant sub-population with a higher MIC, it is advised that the antibiotic exposure should be maintained above the MSW [102,103]. This concept found good application in the study of Zhang et al. [120] concerning danofloxacin in pigs. When the AUC/MPC ratio was equal to at least 24 h, it predicted an antibiotic dose with a high tendency to suppress resistance. This was achievable when the MPC/MIC ratio is not too high. As an example concerning some fluoroquinolones, their MPC/MIC ratios ranged from 5.6 (ciprofloxacin) to 12.2 (sarafloxacin) for *E. coli*, while for *S. aureus*, the ratio ranged from 9 (marbofloxacin) to 136 (difloxacin) [121].

Similarly, the MPC/MIC90 ratio for ciprofloxacin, difloxacin, enrofloxacin, marbofloxacin, and orbifloxacin for *S. pseudintermedius* isolates from dogs was in a narrow range from 5.3 to 6.8 [122]. Based on these results, it is possible to conclude that only the highest doses within the labeled recommended dose ranges of ciprofloxacin (20 mg/kg), enrofloxacin (20 mg/kg), and marbofloxacin (5 mg/kg) could minimize the selection of resistant mutants in vitro [122].

The concept of MSW is currently applied only to fluoroquinolones, likely due to resistance developing through mutational alterations of the drug target. This approach is not applicable to other resistance mechanisms, such as plasmid-mediated, and therefore to other antibiotic classes [123]. Nevertheless, MPC/MIC ratios have been calculated and proposed for several families of veterinary antibiotics, including macrolides, cephalosporins, and florfenicol [124].

Considering the duration of the therapy, thus the time of exposure of bacteria to the antibiotic, Tam et al. [125] found that, to prevent the emergence of resistance for *S. aureus* exposed to garenoxacin, an AUC/MIC ratio of 100 is needed when the exposure time is two days, but the AUC/MIC ratio increases to 280 for an exposure of 10 days. The so-called “one-shot therapy” is a veterinary option that seeks to minimize treatment duration to prevent the emergence of resistance for quinolones by administering high doses of the drug. The goal is to kill the target pathogens as rapidly as possible, allowing the host’s natural defenses to eradicate the remaining bacterial population.

Sumi et al. [103] conducted a systematic review to better understand the antibiotic exposure required to minimize the spread of resistance for Gram-negative bacteria. The review included 56 preclinical studies (in vitro and in vivo) and 2 clinical studies. Results and overall authors’ considerations are summarized below.

(1)The exposures needed to suppress the emergence of resistance for Gram-negative bacteria varied depending on: the antibiotic assayed; the extent of the experiment; the bacterial species and the specific bacterial isolate tested (genomic differences between laboratory reference strains and the corresponding clinical isolates have to be taken into account [126,127]); the bacterial load (the possibility of a pre-existing resistant subpopulation increases in the case of larger bacterial burden [128]); and the PD indices (MIC or MPC) [125,129].(2)As is described in more detail below, patient illness severity should be responsible for the high interpatient variability in antibiotic pharmacokinetics. Commonly, patients in Intensive Care Units (ICU) undergo important pathophysiological changes that have an impact on both the PD and mostly PK characteristics of antibiotics.(3)In vivo studies describing the emergence of AMR in anatomical sites different from the infected one (e.g., gastrointestinal microbiota) were not comprised by the systematic review [130,131].

In conclusion, Sumi et al. [103] suggest that, to suppress the emergence of resistance to Gram-negative bacteria during an antibiotic administration, higher dosages than those usually recommended are needed. This approach must carefully take into account the potential risks of antibiotic toxicity. Additionally, the lack of an immune response in in vitro studies currently limits the clinical application of these attempts.

### 4.2. Patients in Intensive Care Units (ICUs)

Patients (humans or animals, e.g., dogs and cats) in Intensive Care Units (ICU) differ significantly from other patients due to their very serious organic conditions, frequent excretory organ deficiency, high complexity of drug regimens, and severe infection frequently caused by MDR-bacteria, often contracted at the ICU [132,133]. For these patients, antibiotic therapy is initially empirical and revised when the results of bacteriological susceptibility become available. Generally, potential drug–drug interactions (pDDIs) are not taken into consideration. pDDIs may arise from interference with pharmacokinetic processes (these are the more frequent), or they may be pharmacodynamic-based [132].

In ICU patients, managing an antibiotic dosage regimen is a challenge due to the pathophysiological and/or iatrogenic factors that must be considered for the significant effects they may primarily have on kinetic processes. The main consequences are summarized below [132].

(1)Infections are responsible for inflammation processes, which can increase capillary permeability, resulting in an increase of fluids in the interstitial space and a dilution of the systemic concentration of antibiotics. Antibiotics with a low volume of distribution (<20 L/kg bw, β-lactams, aminoglycosides) may especially result in under-dosing [134].(2)Hypoalbuminemia (humans: serum albumin concentrations < 2 g/dL) is frequently observed (humans: 35–40%) and should be taken into account when the antibiotic is highly bound (>80%), as this can lead to a decrease in the total plasma concentration [135,136]. However, this does not necessarily require an obligatory dosage regimen adaptation [104].(3)Altered renal function, or augmented renal clearance (ARC), can result in an enhanced excretion of antibiotics mostly eliminated by the kidney (e.g., hydrophilic β-lactams) [102,137,138].

The inhibition or induction of drug metabolizing enzymes represent the most frequent drug–drug interactions in the ICU [139,140]. The cytochrome P450 (CYP) family is the most implicated in this phenomenon, particularly with the isoforms CYP3A, CYP2D6, and CYP2B6 [141]. Moreover, outstanding attention has been given to transmembrane proteins (e.g., P-glycoprotein, breast cancer resistance protein, etc.) acting as carriers for drugs [142,143].

## 5. AMR in Animals: A Really Complex Phenomenon

The previous section highlighted the importance of the PK/PD approach, which is a pivotal aid in defining the “best” dose/dose regimen to administer to patients with a high probability of success, particularly for antibiotics. However, there are also factors, situations, or differences between antibiotics that make PK/PD underperform, especially in counteracting AMR. The following section provides examples, peculiar to veterinary medicine, to illustrate that AMR is a complex and threatening phenomenon that affects all ecosystems. AMR can only be addressed and contained by a One Health vision.

### 5.1. Aquatic Environments and Aquaculture

Most antibiotic molecules are excreted unchanged and/or metabolized via feces and urine by humans or animals [110,144] and enter wastewater from different sources, as underlined above (see Section 2).

Concentrations of antibiotics in both aqueous and solid-phase media (e.g., biota and sediments) are of the order of “ng/L and μg/L” and “ng/g and μg/g”, respectively [145]. Antibiotic residues in marine ecosystems exert pressure on the bacterial population, leading to the emergence of ARBs and spreading ARGs [146,147].

The research of Maghsodian et al. [148] intended to review the presence of antibiotics in different parts of marine environments (i.e., water and sediments) and in organisms in different aquatic environments (i.e., worldwide seas and rivers; lakes, mainly referring to China). Most of the studies included were carried out in 2018 (15%) and 2014 (11%), and the highest number of research pertained to the Asian continent. The concentrations of antibiotics found in aqueous media were <1 ng/L–100 μg/L. The highest level in water and sea sediments was related to fluoroquinolones [148]. Regarding sediments, the high amounts of quinolones found in the Chinese area can be ascribed to their chelation with cations, metals, and heavy metals, which are generally abundant in sediments (especially sea and lakes) facilitating AMR and the accumulation and propagation of ARGs. Fluoroquinolones (with an abundance of 34%) had the highest average concentration (369.74 ng/L) in Chinese lakes, where they bind to particulate matter. The concentrations of antibiotics in lake water were higher than in rivers and seas [149,150,151]. Sulfonamides showed the highest abundance in rivers worldwide (30%; 191.11 ng/L average concentration), probably related to their widespread use and relatively high stability (one year) [148].

The entry of antibiotics into aquatic environments from different sources may cause their accumulation and magnification in marine organisms (fish, mollusks, crabs, and shrimps). In general, the most frequently prescribed antibiotic classes in aquaculture, including amphenicols, or quinolones (enrofloxacin, nalidixic acid, ofloxacin), tetracyclines (doxycycline, chlortetracycline, and especially oxytetracycline), and sulfonamides (sulfamethoxazole and sulfamethoxazole/trimethoprim combination) all express high chemical stability in water. Conversely, penicillins are easily decomposed in the environment [144,152,153,154,155,156]. Among the different considered antibiotics, the concentrations of fluoroquinolones were higher than those of other antibiotics (68,000 ng/g, with a frequency of 39%). It was concluded that sulfonamides and fluoroquinolones had the highest concentrations in most of the studied environments [148].

Worldwide, over two million tons of fish are produced annually. Although the main producing countries are Norway and Chile, the consumption of fish is increasing in the EU, USA, and Asia [152]. Aquaculture is practiced in a variety of environmental compartments and settings (e.g., in the marine environment, freshwater environment, ponds, nets, closed, open, and recirculating systems), and the impacts could be very different [157].

As previously mentioned, approximately 80% of the antimicrobials used in aquaculture enter the environment in an active form [158]. This can lead to the environment and wild fauna acting as reservoirs of resistance, reintroducing resistant bacteria and ARGs into the food-producing animal and human reservoirs. This resistant material can include commensal or pathogenic microorganisms, such as *Salmonella* spp. and *Vibrio* spp. [152]. Another important topic is the application of sludge (e.g., salmon sludge) to agricultural soils, which, while having positive aspects of fertilization, can also represent a potential source of further ARG environmental diffusion [159]. Aquatic beings, as well as terrestrial ones, can be considered as entropic systems, where intestinal microorganisms interact with environmental ones, resulting in “ARGs trade”. In this “ARGs life-cycle”, the final human consumer microbiome could also be impacted by the commensal strains in seafood, which may harbor resistance genes. The above conditions are realistically responsible for the emergence and spread of multidrug-resistant (MDR) or pan-resistant pathogenic or commensal microorganisms [160]. Furthermore, two other points should be taken into consideration: (i) several studies have indicated a statistically significant correlation between high water temperature (simulating the “global warming” phenomenon) and infected treated finfish, suggesting that warm water could be responsible for increased AMR diffusion; (ii) a different high direct correlation was demonstrated between heavy metal residues (such as lead, cadmium, and mercury) and the horizontal ARG transmission (for example, more specifically observed for tetracycline molecules: *tet* genes) [152,161].

### 5.2. Intestinal Microbiome and Antibiotics MRLs in Food of Animal Origin

#### 5.2.1. Intestinal Microbiome

The human intestinal microbiome is characterized by a complex group of microorganisms composed of bacteria, fungi, archaea, protozoa, and viruses. These microorganism are arranged in a community living in close relationship with the human GI tract environment and representing a good habitat for bacteria to exchange genetic materials (i.e., plasmids), comprising ARGs [162].

The commensal microbiota of the gastrointestinal tract or the skin of animals, as well as of humans, could be involved in AMR, and this is significant also from an ecological perspective. In animals, commensal microbiota are characterized by large biomasses (expressed in tons, rather than mg for the target pathogen or kg for the commensal microbiota) that significantly outweigh the biomass of pathogens harbored by animals [28]. For example, in a cow with pasteurellosis, the total lung pathogen load is at most a few mg, while the bacterial mass of the corresponding commensal microbiota harbored by the same animal is several kg. Consequently, the potential risk when treating a pulmonary infection is the exposure of the intestinal microbiota to the antibiotic. The commensal microbiota bacteria are not pathogenic, but they harbor, even before any antibiotic administration, a range of genes of resistance, known as the resistome [28]. The use of veterinary antibiotics can promote the selection and amplification of this pool of genes, which may be transmitted to other organisms (animals or humans). Moreover, the intestinal microbiota is regularly excreted into the environment through fecal emission at a high rate, leading to the spread of bacteria, including those harboring genes of resistance. In cattle, feces production rates are 12 kg/day for calves, 26 kg/day for beef, and 62 kg/day for milk cows. In pigs, the daily production of manure is 1 to 4 kg, and for egg-laying poultry, it is approximately 100 g. This is by far the largest connection route for the transfer of resistant bacteria and gene elimination between the animal and the human resistome [28].

#### 5.2.2. Residues of Antibiotics in Food of Animal Origin

Drug use is pivotal to treat diseases in food-producing animals; it is estimated that 80% of global terrestrial and aquatic livestock receive medication at least once during their lifetime. Similarly to any other drug, the use of antibiotics in food animals and in aquaculture can result in the presence of residues in edible tissues, as well as in milk and honey [163]. Such as other authorities worldwide, the European Union has set acceptable daily intake values (ADIs) and maximum residue limits (MRLs) for veterinary drugs (Regulation 470/2019) [164]. The ADI considers human toxicological and microbiological endpoints, which are related to intestinal dysbiosis caused by antibiotic concentrations exceeding the MIC for human enteric bacteria, and the selection of resistant bacteria within a susceptible population [163]. Recently, it has been demonstrated that antibiotic concentrations well below the MIC can select for resistant bacteria within communities of susceptible bacteria [165]. This is of great concern, and thus it is important to take into consideration the minimum selective concentration (MSC), namely the lowest concentration of an antibiotic that provides a strain of resistant bacteria a selective advantage over susceptible strains [166]. There are two components of the MSC: (i) the minimum concentration at which de novo resistance can be induced (MSC*_denovo_*), and (ii) the lowest concentration that selects for a resistant, compared to a susceptible, strain (MSC*_select_*) [165,167].

The results of the study [163] demonstrated that, based on the ADI values of tetracycline, oxytetracycline, ciprofloxacin, sarafloxacin, erythromycin, spiramycin, tilmicosin, tylosin, and lincomycin, the residue concentrations (MRLs) in human colon may be up to 1000-fold higher than their MSCs (tetracycline, oxytetracycline, and tilmicosin: 1000-fold higher; other antibiotics: about 10-fold higher than the upper limits of their MSCs). In conclusion, the observed colon concentrations of the considered antibiotics, corresponding to MRLs and hundredfold below the MICs, are able to select for resistant bacteria and de novo resistant mutants and to promote horizontal gene transfer [165,166,168].

Bringing one more example, Gonzáles et al. [169] hypothesized that the low concentrations of ciprofloxacin used in animals and found as residue in meat may play a role in the genesis of quinolone resistance in *Neisseria gonorrhoeae*. The MSC*_denovo_* (0.004 μg/L, 1/1000 of the MIC) was considerably lower than the quinolone concentrations found in foodstuffs. Even though a number of factors can affect the MSC, the authors affirmed that ciprofloxacin and quinolone concentrations in meat may be linked to the prevalence of ciprofloxacin resistance in *N. gonorrhoeae*, suggesting that it could be positively associated with quinolone consumption for animal husbandry [78,170,171].

In assessing whether antibiotic residues equal to MRL in products of animal origin may facilitate AMR, several factors and variables should be taken into consideration: the combination of different antibiotics present in a particular foodstuff (e.g., meat and milk), the model of diet (e.g., geographical area, individual socioeconomic status, culture), and the cooking method used (e.g., boiling can reduce macrolide and quinolone residues in meat by more than 75%) [163,165,172]. Depending on the chemical properties of the molecules, their stability against heating during cooking can vary, although there are certain discrepancies among researchers. While some antibiotics tend to be heat stable (quinolones, nitroimidazoles, nitrofurans), others tend to be more heat labile (tetracyclines, macrolides, sulfonamides, ß-lactams, and aminoglycosides). In conclusion, significant variations can be observed for the same drug between different heat treatments [77].

Antibiotic residues in foods of animal origin are not as much of a direct public health concern; however, the long-term exposure of human intestinal microbiota to antimicrobial MSCs can favor the selection of resistant bacteria and enable the transfer of resistance genes in the human gut [163,173].

### 5.3. Extra-Label Use

In veterinary medicine, using drugs, and specifically antibiotics, in a way not in accordance with approved label directions (the so-called extra-label or off-label use) is frequent, and it is more common in companion animals than in food-producing species. Papich [88] described the prescribing practices of antimicrobial agents in companion animals in the United States. In the U.S., the extra-label administration of drugs, and specifically of antibiotics, in companion species is legal, it is legal to administer human-labeled antibiotics, and it is legal to use products licensed for other species (e.g., food-producing animals). Currently, the number of antibiotic products approved by the Food and Drug Administration (FDA) for dogs and cats is very limited [174].

Many medicinal products are no longer marketed or available in all forms in the U.S. The indications listed on the label are generic and not relevant (e.g., “for treatment of susceptible bacteria in dogs and cats”) or outdated, as well as dose regimens are not sufficient and accurate according to current PK/PD principles. Administering antibiotics that are not active against the bacteria listed on the label, or when the dose on the label is insufficient (due to AMR), results in unnecessary antibiotic exposure [88]. Most of the above considerations related to the United States accurately depict the situation and are similar in other countries, although there are significant differences in regulations or legislations (e.g., in Europe, where the extra-label use of drugs is strictly regulated).

The Guidelines for the Prudent Use of Antimicrobials in Veterinary Medicine issued by the European Commission in 2015 state: “The off-label (cascade) use of antimicrobials not authorized in veterinary medicine to treat non-food-producing animals should be avoided, especially when the drugs are of critical importance for human health (e.g., carbapenems and tigecy-cline). Their use should only be considered in very exceptional cases, for example, when laboratory susceptibility testing has confirmed that no other antimicrobials will be effective and where there are ethical reasons to justify such a course of treatment” [175].

The extra-label use of drugs is allowed “in particular situations”, meaning that a medicinal product can be exceptionally administered for non-authorized use in specific cases (e.g., to avoid suffering to animals) under the veterinarian’s responsibility. This practice is well regulated by authorities (for Europe the Regulation 2019/6 [13]) and involves using the product in a species not listed on the label, at a different dosage than described, or to treat a condition not indicated [176].

The procedures for the proper extra-label use of drugs in food animals are significantly more restrictive than those for companion animals [77]. For example, concerning antibiotics in aquaculture, the prescription under the “cascade” allows the veterinarian to prescribe veterinary drugs authorized for different target species (e.g., pigs) when there is no alternative available for fish [157]. This exceptional prescription is beneficial for animal health and welfare, but it is associated with risks (e.g., treatment failure due to incorrect dosage regimen, toxic effects, increased selective pressure), since the initial therapeutic protocol and environmental impact have not been evaluated for that off-label use [157].

#### 5.3.1. Dose Extrapolation

Safe and effective drug dosing is essential, regardless of its purpose of administration. As the PK/PD approach applied to antibiotic therapy emphasizes, for such drugs, the “correct” dose is pivotal. When antibiotics are underdosed, the patient does not recover, and the risk of creating and spreading bacterial AMR is significantly increased.

The dose extrapolation of drugs based on body weight (mg/kg) alone is not the most accurate approach. The relationship between the metabolic rate of an animal and its size must be taken into consideration. As the size of animal species increases, the body surface area in relation to body weight decreases, and all the physiological processes of a larger animal are slower than those of a smaller animal (e.g., heart rate, respiration rate, movements, etc.). In this context, the “metabolic rate” (closely related to body surface area) provides an index of how the rates of all the physiological processes involved in drug absorption, distribution, metabolism, and excretion scale across different species [177,178].

The application of this principle into practice, and thus into the need to extrapolate the dosage of a drug from one animal species to another (e.g., extra-label use), is based on the general consideration that the higher the body weight, the lower the doses of drugs (mg/kg). It is well known that the final therapeutic effect of a drug depends on its pharmacokinetic (absorption, distribution, metabolism, and excretion) and pharmacodynamic (affinity for the target, the nature and magnitude of the response produced by the drug–target interaction) processes. Some of these are related to the body surface area of the animal, while others are not (Table 3).

Sharma et al. [177] and Nair and Jacob. [178] provided principles, suggestions, or methods of inter-species dose extrapolation. However, when antibiotics are used, it is essential to consider the global therapeutic protocol (administration route, intervals between administrations, duration of therapy, etc.) and to bear in mind that in each animal species breed, animal age, sex, and body condition, as well as the specific disease (e.g., bacterial urinary infections or bacterial bile duct infections), could affect the disposition of the drug in plasma.

#### 5.3.2. Pharmaceutical Dosage Form

Problems in the use of extra-label drugs are related to dosage forms, such as the concentration of the active ingredient, the type of preparation (solid or liquid), the size of solid forms (i.e., tablets) or the possibility to divide them, the type of excipients, and the organoleptic characteristics (taste or odor). These are just some of the many factors that must be taken into consideration when, especially in the case of antibiotics, a veterinary medicinal product labeled for a given species is used in a different one [88].

### 5.4. Exotic Pets and Wildlife

#### 5.4.1. Exotic Pets

To control the spread of AMR, action plans, recommendations, and guidelines on the responsible and prudent use of antibiotics were developed for food-producing animals and companion animals, but they appear to be lacking in exotic pets. This is likely due to the scarcity and/or lack of data and information on the use, overuse, or misuse of antibiotics in these species [14].

First of all, most of the monitoring programs to assess the quantity of antibiotics used in animals are focused on food-producing animals, while data on companion animals (e.g., dogs, cats, horses, and rabbits) are scarce, and there is no information available on exotic pets [14]. In the literature, only one published study was found that provided data on the use of antibiotics in such species. The study is not exhaustive and only provides indications that most antimicrobial prescriptions are unnecessary or inappropriate and that most antimicrobial prescriptions seem to be empiric and not based on proper diagnosis. No information was produced regarding the amount or type of antibiotics used [14].

MDR bacteria, such as methicillin-resistant staphylococci, and extended-spectrum β-lactamase-producing *Enterobacteriaceae* have been signaled in birds and rabbits, as well as in birds and turtles, respectively [179,180,181,182].

In addition to the lack of information on the use of antibiotics in exotic pets, it should be noted that the possibilities of choosing among the available antibiotics for the different species are limited (Table 4). Beyond instances synthetized in Table 4, several other factors make antibacterial treatment in exotic pets difficult: (i) proper pharmaceutical formulation (liquids are preferred over tablets); (ii) concentration; (iii) taste of the formulation; (iv) availability of scientific literature regarding the dosing regimen of antibiotics in exotic pets; (v) information on bacterial pathogens located in a specific area of the organism in the different species; (vi) the potential species-specific toxic effects [14].

The aforementioned challenges direct the choice towards veterinary medicinal products authorized for different animal species or for human use, compromising the selection of the right antibacterial in a specific bacterial infection in a particular patient. However, as with food-producing animals and companion animals, antibiotic treatments in exotic pets must be settled according to the responsible and prudent use approach.

#### 5.4.2. Wildlife

As previously mentioned, the use of drugs in food-producing animals and drug residues in tissues can affect wildlife and the environment. Drug residues have been detected in wild animals even when they were never treated. A high percentage of samples (29%) taken from carrion disposed of for feeding to endangered scavenger birds had antibiotic residues, with oxytetracycline (at the highest concentration: 1452.68 ng/g) and trimethoprim being the most common [186].

Fluoroquinolones (marbofloxacin, enrofloxacin, ciprofloxacin) and nafcillin were detected in vulture plasma. Enrofloxacin and marbofloxacin (up to 20 µg/L, and ∼150 µg/L respectively) were quantified in a high proportion of individuals (92%) in different colonies and on different dates [187]. Similarly, enrofloxacin (54.5 ± 6.6 µg/L) was found in the plasma of nestlings of a golden eagle (*Aquila chrysaetos*) [188].

The detection of MDR *Staphylococci* strains in tawny owls, scops owls, and black kite owls [189] proves the widespread diffusion of resistant bacteria in the environment, and it must be considered that wild birds contaminate several different habitats with their droppings, representing a source of pathogens and antibiotic-resistant bacteria [190,191]. Another instance of the contribution brought by wildlife to the spreading of AMR is that of Cagnoli et al. [192]. The authors investigated the possible involvement of wild central Italy avifauna (raptors, synanthropic birds, aquatic birds) in the spreading of antibiotic-resistant enterococci. *Enterococci* (genus *Enterococcus*) are Gram-positive opportunistic pathogens that are part of the gastrointestinal microbiota of humans and animals. Alongside positive effects to the host (i.e., probiotic activity, bacteriocin production), occasionally they act as pathogens and are able to acquire and transfer genes for AMR to different molecules [193,194,195,196,197,198,199,200]. The results obtained confirmed that AMR is widely spread among *Enterococcus* strains. All but one of the isolates were multi-drug-resistant (MDR); about 20% were categorized as extensively drug-resistant (XDR), and 3% was possibly pan-drug-resistant (PDR). Most of the assayed antibiotics were effective against less than 50% of isolates, and none of the antibiotics were effective against all the isolates. The high and diffuse resistances observed, especially for aminoglycosides and tetracyclines, were particularly notable. The highest percentage of *Enterococcus* isolates showed resistance to fluoroquinolones (70.87% to enrofloxacin and 51.46% to ciprofloxacin). The observed lower susceptibility to fluoroquinolones in enterococci from wild birds confirmed data described by other authors in raptors, common buzzards, and wild birds [195,196,197,199]. Quinolones, and especially fluoroquinolones, are frequently used in livestock and in companion animals [18], and as previously pointed out, they are frequently detected in the environment, in particular in wastewater and surface water [201].

The findings of Cagnoli et al. [192] pointed out that wild birds may behave as a “reservoir” and source of ARGs and ARBs for humans and domestic animals, as well as synanthropic birds and migratory birds (mainly waterfowl). Lastly, hunted waterfowl could directly transfer ARBs to humans when processing carcasses.

## 6. Conclusions

Antibiotics are among the most important treatment options available in human and veterinary medicine, but AMR is a serious public health concern. This narrative review seeks to provide an overview of the state of AMR in veterinary medicine, considering both the advantages and disadvantages of using antibiotics in animals in relation to this global issue. The *fil rouge* of the review is the close relationship that exists between humans and animals, which is rooted in the fact that, despite the differences in species, both are complex organisms with similar biological processes. Starting from AMR, the authors focused on the use of antibiotics in animals, emphasizing the importance of responsible and prudent use, and the need for the veterinarian to be aware of the PK/PD approaches that can optimize dosage regimens, even in critically ill patients (ICU).

By citing specific instances (aquaculture, extra-label use, antibiotic MRLs, exotic/wild animals), the authors intended to emphasize the complexity and resulting critical issues of the use of antibiotics in animals, which, particularly in food-producing species, has had and still has remarkable responsibilities. In this context, the veterinarian plays a key role, being called to carefully evaluate the consequences of its decisions and having the liability to educate everyone who administers antibiotics to animals (e.g., breeders and pet owners).

Antimicrobial stewardship involves coordinated approaches and interventions. It not only focuses on reducing antibiotic use, but it also encompasses infection control, clinical microbiology, the surveillance of antibiotic use and AMR, as well as pharmacovigilance, education, guidelines, and regulations. Most veterinarians are aware of AMR, but veterinary antimicrobial stewardship is a relatively new concept in veterinary medicine that needs to be further consolidated and pursued. However, as this narrative review clearly highlights, a shared commitment between the animal and human medicine worlds is an added value in the fight against AMR and represents an effective application of the One Health approach.

## Figures and Tables

**Table 1 antibiotics-12-00487-t001:** MDR bacteria isolated from dogs and cats.

MDR Bacteria	References
Methicillin-resistant *Staphylococcus aureus* (MRSA)	[45,46]
Methicillin-resistant *Staphylococcus pseudintermedius* (MRSP)	[45,46]
Vancomycin-resistant *Enterococci* (VRE)	[47,48]
*Escherichia coli*-extended-spectrum beta-lactamase (ESBL)	[47,49,50,51]
AmpC and carbapenemase-producing Gram-negative bacteria:	
*Escherichia coli*, *Klebsiella pneumoniae*	[47,49,50,51]
*Pseudomonas aeruginosa*	[52]
*Acinetobacter baumannii*	[53]

**Table 2 antibiotics-12-00487-t002:** Main conditions to ensure the proper prescription of veterinary antibiotics [17].

Prescribing after clinical examination; Preventive use of antibiotics should be avoided or restricted to particular conditions, especially in food-producing animals; Avoiding off-label use or, when not possible, adapting the dosage regimen based on scientific considerations; Avoiding the use of CIAs, except for patients showing critical conditions; The prescription of CIAs must be justified by the results of susceptibility tests.

**Table 3 antibiotics-12-00487-t003:** The main determinants of pharmacokinetics and pharmacodynamics related or not to the body surface area [177].

Process	Correlation
*Pharmacokinetics*	
Absorption	Mainly influenced by drug physicochemical properties; Membrane permeability is independent of size, with exceptions related to first-pass metabolism and transporters such as P-glycoprotein.
Distribution	Blood flow and rate of diffusion to target cells are related to size.
Protein binding	Independent of size (species-specific differences).
Transporters (P-glycoproteins)	Independent of size (species-specific differences).
Hepatic metabolism	Blood flow and rate of diffusion to target cells are related to size + independent of size (species-specific differences in enzymatic families).
Elimination	Blood flow and rate of diffusion to target cells are related to size + independent of size (species-specific differences).
*Pharmacodynamics*	Independent of size (species-specific differences).

**Table 4 antibiotics-12-00487-t004:** Main instances concerning formulations of antibiotics for exotic pets.

Exotic Pets	Instances
Birds	Liquid oral forms of aminopenicillins for human use are suitable (concentration of active principle; acceptable taste) [14].
Small birds and rodents	Azythromycin liquid formula for human use are suitable [183].
Rabbits, guinea pigs, other herbivorous	Limited treatment options because oral administration of antibiotics may result in dysbiosis and enterotoxaemia [184,185].
Reptiles	Ceftazidime (third-generation cephalosporin authorized for human use only, and classified as HPCIA) is specific for *Pseudomonas* spp. infections [14].

## Data Availability

No new data were created or analyzed in this study. Data sharing is not applicable to this article.

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
