# Peer review of "The Use of Antibiotics and Antimicrobial Resistance in Veterinary Medicine, a Complex Phenomenon: A Narrative Review"

_antibiotics, 2023, doi:10.3390/antibiotics12030487_

Round 1

Reviewer 1 Report

Dear Editor,

Thank you for the provided opportunity to review the manuscript by Caneschi et al. on PK/PD approach in veterinary medicine.

The manuscript is unfocused. While Pk/PD approach is stressed in the title, the aim of the study was different - AMR in veterinary medicine, including different animal species and the environment. In conclusion, veterinary stewardship and specific instances for the application of antibiotics are described.

In the manuscript, both the application of antibiotics in veterinary medicine and medicine are described with human AMR being prevailing in comparison with veterinary medicine.

Another concern is that the authors extensively describe the residues accumulation in the environment that is far from PD/PK, the area of the research of the present manuscript.

Among animals, the companion animals were analyzed, however, the productive animals are usually considered to be the main drivers of antimicrobial resistance.

The above-mentioned concerns must be addressed to proceed further with the manuscript.

Other comments:

The authors quoted the R2019/6, however, this is not needed.

Line 80-82. Not clear.

Line 85. Please provide references for "many studies and research"

Please explain PK/PD approach in the introduction.

Line 100. "Any". Also justified and according to the prescription?

Line 102. "Selection pressure". Selective pressure?

Line 103. Skin flora? Please check usage of "microflora" and "flora" and change to "microbiota" throughout the manuscript

Line 111. What is "low-level resistance"?

Line 120. Provide the appropriate references for those "several studies"

Line 124. The intestinal tract is not the source of ARGs, I guess

Line 143. Please provide a reference for the application pattern of antimicrobials. Families or classes?

Line 150. ""Critical.... ". Terminology! Please explain how resistance to tet is associated with MDR?

2.1. Food-producing animals. Only pigs and chickens?

Line 157. The authors describe that the ARGs are associated with feces. The author repeats this in every section. Are there other mechanisms/ materials related to the spread of AMR? Can the authors explain in general the spread of AMR in animal populations?

2.2. "emerged at lesser extend". Has the resistance also emerged in productive animals? Still not clear.

Table 1 is a simple list of resistant microorganisms isolated from companion animals.

Line 174. These? Please explain.

Line 174-183. Children, elderly - is this about pets? MDR is dangerous as a life-threatening condition in hospitalized human patients.

Line 184-195. Is this paragraph about one study in China?

Line 185. What is "ESBL"?

Line 189, 190, 191. Please provide appropriate references.

Line 194. How were the isolates compared?

Line 199. ARG spreading ability?

Line 213. Highest .... Antibiotics". Is this the term stated by WHO?

Line 215-235. The authors must avoid excessive quoting, the readers can find and read this themselves.

Line 248. Avoiding HPCIAa. Avoiding at all? Is this a review on medicine?

Section 3 is boring to read. If the authors prefer to focus on companion animals, they must state it clearly in the aims of the study. I am not sure about the relevance and interest of lines 300-334 for readers.

What are "veterinary CIAs"?

Line 275-276. Unclear.

Line 280. Please provide a reference for growth promoters.

Line 283-290. Please check, the authors had recalculated their antibiotic consumption prognosis in the recent study.

Lines 348-353. Common but representing only 1%.

Line 413. Were the conclusions drawn from one study?

Line 427-430. Unclear. Provide the appropriate references.

Line 466-494. Please do not rewrite the findings of previous reviews.

Line 506-527. Is this about humans?

4.4. subsection. Why do the authors describe MCM instead of describing modeling for the demonstration of PK/PD?

Could the authors provide an overview of vet studies on the evaluation of PK/PD?

Line 569. In all lakes around the globe?

Line 573-584. Is this about the research in a known country?

Line 602. Could the authors provide some examples of contamination of feed with ARGs?

These conditions. Which conditions?

Section 5. describes the problem of the residue in veterinary medicine, which is different from the AMR or PK/PD approach. 5.2. subsection is dealing with human microbiome which is not relevant to the present study. The extra label usage of antibiotics had been already explained within the USA context.

Tables without columns?

Line 743-749. Please provide some examples.

Line 802-828. This all is about feces, dropping, and AMR but not about PK/PD stated in the title of the manuscript.

Conclusions are not supported by the review findings.

Reviewer 2 Report

This is a well-written narrative review of a current and clinically relevant topic. The main broad topics concerning antibiotic use in veterinary medicine are touched on, but given the nature of the subject, the level of depth is lacking for some of these topics. For example, the section on exotic animals and wildlife potentially warrants a separate review of its own to cover sufficient breadth and depth of the topic. The other deficiencies of the manuscript are related to the nature of a narrative review as opposed to other forms of reviews, such as scoping and systematic reviews, e.g. there is no methodology described to enable replication in future when further research has been done; risk of bias due to the lack of consistent methodology for discovery of related literature; lack of comprehensiveness compared to the robust methods described for systematic reviews. The type of review this manuscript is should be more obvious in the title, abstract and introduction of the manuscript that this is a narrative review, so that the expectations of readers are managed.

Round 2

Reviewer 1 Report

Dear Authors,

Thank you for your responses and amendments according to my points raised. The manuscript had been improved significantly. Some additional points to be addressed:

Line 66-71. Rephrase this paragraph in your own words, ie. the EU Commision has highlighted the the cross-border dimension, economic impact, and animal and human health consequences that requires global intersectoral action to combat AMR

Line 102. Why are the experts put in quotation marks? I also could not understand "also everyone else who". Person who has the right to prescribe antibiotics?

Line 113. Please define "These genetic elements"

Line 129. Explanation for abbreviation ARG is needed.

Line 134. Please change "antibiotics" to "antimicrobials" according to the WHO terminology

Line 53. Please change "microflora" to "microbiota"

Table 1. Please consider an addition of following column  - country/ year of isolation, place of isolation (ie. veterinary setting), replace "study" with references

Line 235. Please replace "flora" with "biota".

4.4. subsection does not contain enough information for separate subsection, please consider merging with previous section

Lines 538-545. Could the authors include the counties in which the described research was/ were conducted?

Line 548-554. Please shorten this sentence as "the most frequently used antibiotics included... all expressed high chemical stability in water".

Line 594. "both these large" replace with "commensal microbiota"

Line 600, 604. "Flora" replace with "microbiota"

5.3. subsection. Explain meaning of "extra-label use"

Line 678-684. Could the authors rewrite this paragraph in their own words? This also will help to use the same terminology (off-label vs. extra-label).

Are there any consequences of application of the extra-label antimicrobials for food animals in the EU?

Table 3. Title. Please add "of pharmacokinetics and pharmacodynamics". The titles of columns are missions.

5.3.2. Could this section be merged with the previous?

5.4.1. Could the authors describe which animals are exotic in this review? IE. wild birds are mentioned both in subsections 5.4.1. and 5.4.2.

Line 748. Could the author provide some examples of animal species that have been treated empirically?

Table 4. Please add the titles for each column. The second column may be further split into animal species/ antibiotic/mode of application/ reference

Line 771. Please provide the appropriate reference after "never treated".

Line 786. Please replace "microbiota" with "microbiota"

Line 823. Antimicrobial stewardship needs to be discussed in the manuscript
